# Optimizing the Thermomechanical Process of Nickel-Based ODS Superalloys by an Efficient Method

**DOI:** 10.3390/ma15124087

**Published:** 2022-06-09

**Authors:** Wuqiang He, Feng Liu, Liming Tan, Zhihui Tian, Zijun Qin, Lan Huang, Xiangyou Xiao, Guowei Wang, Pan Chen, Baogang Liu

**Affiliations:** 1State Key Laboratory of Powder Metallurgy, Central South University, Changsha 410083, China; 183301019@csu.edu.cn (W.H.); liufeng@csu.edu.cn (F.L.); limingtan@csu.edu.cn (L.T.); zijun.qin@csu.edu.cn (Z.Q.); lhuang@csu.edu.cn (L.H.); xiangyou.xiao@csu.edu.cn (X.X.); wangguowei23@csu.edu.cn (G.W.); 2Research Institute of Powder Metallurgy, Central South University, Changsha 410083, China; 3College of Science, Huazhong Agriculture University, Wuhan 430070, China; 2017301210207@webmail.hzau.edu.cn; 4School of Minerals Processing and Bioengineering, Central South University, Changsha 410083, China; 5School of Energy and Electromechanical Engineering, Hunan University of Humanities, Science and Technology, Loudi 417000, China

**Keywords:** nickel-based ODS superalloys, consolidation, strengthening, hardness

## Abstract

Thermo-mechanical process of nickel-based oxide dispersion strengthened (ODS) superalloys is critical to produce desired components. In this study, an efficient method of consolidating powder is introduced to optimize the preparation process, microstructure and properties of nickel-based ODS superalloys. The influences of consolidation temperature, strain rate and ball milling time on the hardness of nickel-based superalloys were studied. The relationship among process, microstructure and hardness was established, the nanoparticles strengthening and grain boundary strengthening in nickel-based ODS superalloys were discussed. The results indicate that long ball milling time, moderately low consolidation temperature and high strain rates are beneficial to improving properties of nickel-based superalloys. Moreover, dispersion strengthening of nanoparticles and grain boundary strengthening play important roles in enhancing nickel-based ODS superalloys.

## 1. Introduction

Nickel-based oxide dispersion strengthened (ODS) superalloys have been considered as promising candidate materials such as gas turbines, heat exchanger tubes and nuclear reactors due to their excellent radiation resistance, comprehensive mechanical properties and high temperature creep strength [1,2,3,4,5]. These enhanced properties are mainly attributed to high number density nanoscale particles, which can pin grain boundaries and dislocations [6,7,8]. The addition such as YH_2_ and Y_2_O_3_ is dissociated to Y during the mechanical alloying (MA) process, then Y reacts with and Al or Ti in the matrix to form nanoscale Y-Al-rich or Y-Ti-rich oxides (e.g., YAlO_3_, Y_3_Al_5_O_12_,Y_4_Al_2_O_9_, Y_2_Ti_2_O_7_, Y_2_TiO_5_) in consolidation process [9]. The hot extrusion (HEX) and hot isostatic pressing (HIP) are commonly used consolidation processes to prepare nickel-based ODS superalloys. However, the preparation of ball milling and consolidation of powder usually takes several months and thus it is costly, which limits the development of ODS superalloys.

It is well known that the properties of nickel-based superalloys including nickel-based ODS superalloys are strongly dependent on the processing parameters. For example, Kim et al. [10] investigated the influence of several processing parameters including milling temperature, rotation speed, and consolidation temperature on microstructure and properties of 14Cr ODS steels alloy. As milling temperature decreases, milling speed increases, and HIP temperature decreases, the microstructures of samples become finer and more uniform, and considerable improvement of tensile strength was noticed. Kishimoto et al. [11] and Odette et al. [12,13] reported that with the increase of HIP temperature the grain size of the alloy grew significantly, and the dislocation density decreased, the average particle size of oxides increased and the density of oxides decreased. Tan et al. [14] studied the effects of HIP temperature and pressure on the grain in PM superalloy FGH96, and it was found that temperature and pressure played different roles in controlling prior particle boundary (PPB) precipitation and grain structure during HIP, the tendency of grain coarsening under high temperature could be inhibited by increasing HIP pressure which facilitated the recrystallization. It is essential to optimize the processing parameters to obtain components with desired microstructure and properties. Therefore, to accelerate the development of nickel-based superalloys, some efficient experimental methods need to be developed [15].

In this study, a new efficient thermal consolidation method was used to prepare nickel-based superalloys with and without ODS. It takes only a few minutes and a few grams of powder to prepare a sample by this method, which dramatically shortens the preparation cycle and reduces the cost. In addition, the influences of consolidation temperature, strain rate and ball milling time on the hardness of nickel-based superalloys were studied, and the relationship of process, microstructure and properties was established, the strengthening mechanism of nickel-based superalloys with and without ODS were discussed.

## 2. Materials and Experiment Methods

The chemical composition of two nickel-based powders (PA1 and PA2) prepared by argon atomization in this work was given in Table 1. MA powder was prepared by ball milling PA powders and 0–0.6 wt.% YH_2_ powders for 0–36 h with speed 350 rpm in high-energy planetary mill filled with high purity argon gas (99.999%). The ball-to-powder mass ratio was 8:1 and 0.5 wt.% ethanol was added as process control agent. The MIX powder was prepared by mixing MA and PA powder at ratio 1:2 (wt.%) for 12 h with speed 60 rpm. The laser particle size analyzer (Mastersizer 3000, Malvern Panalytical, Malvern, UK) was adopted to detected the particle size distribution of powder.

The powder was sealed in stainless steel container with 10 mm diameter and 15 mm height, vacuum electron beam welding is adopted to welded container under a vacuum of 10^−2^ torr. Thereafter, the container was compressed by Gleeble 3180D at temperatures ranging from 850 °C to 1150 °C with strain rates 0.1 s^−1^–5 s^−1^. Carbon sheets and lubricant were placed between specimen and dies to make the deformation more uniform during compression. The specimens were originally heated to certain temperatures with a rate of 5 °C/s and held for 3 min to homogenize temperature of specimen. To monitor the temperature during the compression, the thermal-couples were welded at the longitudinal center of the specimen surface. Finally, the specimen was compressed according to the preset procedure and immediately water quenched to freeze the deformation microstructure. All specimens were compressed to a total engineering strain of 80% with constant strain rates. The process flow diagram of specimens prepared is indicated in Figure 1, and the processing methods of specimens have been compared in Table 2.

The phases of powder and alloys were investigated using X-ray diffractometer (XRD, D/MAX 2500, RIGAKU, Tokyo, Japan) with Cu Kα radiation of 0.15405 nm. The diffraction Angle (θ) ranges from 20° to 80° with scanning rate of 1 °/min.

The Vickers microhardness measurements were performed by using vickers hardness tester (THV-10, TEST-TECH, Shanghai, China) at a load of 3000 g and a dwell time of 10 s. The standard deviation was calculated based on 10 measurements.

The characterization of microstructure alloys was observed by optical microscopy (OM, DM4000M, LEICA, Germany) and field-emission SEM (Quanta 650, FEI, Hillsboro, OR, USA) equipped with an electron backscatter diffraction (EBSD), the central of samples was used to analyzed, as shown in Figure 1. The specimens were mechanically polished by abrasive papers and 50 nm aluminium oxide, the specimens for OM observation were etched by a solution of 100 mL ethanol + 100 mL HCl + 5 g CuCl_2_, and the specimens for EBSD observation were vibration polished for 3–8 h. The vibration polishing was conducted on Buehler vibratory polisher VibroMet 2 (Buehler, USA) with 30% amplitude. The EBSD was conducted at 20 kV accelerating voltage, 0.05 or 1 µm step size, the data was analyzed via HKL Channel 5 software and the equivalent grain size was determined based on grain area *A*, $2\sqrt{A/\pi }$.

Transmission electron microscope (TEM, Themis Z 3.2, FEI, USA) observation with energy dispersive spectrometer (EDS) was used to characterize the structure of specimens, under 200 kV accelerating voltage. The TEM samples were polished to a thickness of 50 µm by abrasive papers, and sectioned to diameter 3mm. Then the slices were twin-jet electropolished at −25 °C and 40 V in the corrosive solution of 90% ethanol and 10% perchloric acid.

## 3. Results and Discussion

### 3.1. Morphologies of Powder Analysis

Figure 2a,c show the SEM images of precursor powders, it is evident that PA powder is spherical or subspherical, the EBSD inverse pole figure (IPF), grain distribution and grain misorientation are shown in Figure 2b–f. The grain size ranges from 0.5 µm to 18 µm, the average grain size of PA1 and PA2 powder is 4.8 µm and 5.8 µm, and the average grain misorientation of PA1 and PA2 powder is 33.8 and 31.4, respectively.

The microstructure and particle size distribution of powder with different ball milling time are shown in Figure 3. The particle size of PA powder ranges in 1–170 µm, the powder particle is irregular morphology after ball milling, because of the repeated and high-speed collision between powder and powder or powder and ball milling medium during MA. Large plastic deformation of powder after crushing during the ball milling process. Within ranges of 20–600 µm, the particle size increases with the increasing ball milling time, specifically, the average particle sizes of PA1 and PA2 powder are 182 µm and 224 µm after ball milling 24 h, and these of PA1 and PA2 powder increase to 205 µm and 245 µm after 36 h ball milling. The average particles size of powder increases due to welding of powders during ball milling.

### 3.2. Macroscopic Cracking Analysis

96 specimens were prepared at temperatures ranging from 850 °C to 1150 °C with strain rates of 0.1–5 s^−1^. The shapes of the specimens after hot compression are shown in Figure 4. From macroscopic perspective, most of the specimens are regular pie with obvious bulging in the middle, which is due to the non-uniform deformation caused by the friction between the specimen and workpiece during hot compression [16]. Some specimens with obviously macroscopic failure features (marked by the red box) are observed, these specimens are compressed at the low temperature, the cracking percentage of the samples is 14.6% and 5.2% compressed at 850 °C and 950 °C, respectively. No macroscopic cracking was observed in samples consolidated at 1050 °C and 1150 °C.

Figure 5 depicts some typical features of PA1 and MA1-36h alloys consolidated at different temperatures with strain rate of 0.1 s^−1^. The microscopic cracks along the PPB are observed in PA1 alloys and MA1-36h alloys at 850 °C corresponding to Figure 5a,c, respectively. Generally, PPB are difficult to be broken and eliminated at low temperature, which facilitates the crack nucleation and propagation during thermal consolidation [17]. The microcrack along the PPB are mainly caused by local stress concentration [18,19]. With the increase of consolidation temperature, the thermal activation of the material increases, the atomic diffusion rate accelerates, and the kinetic energy of the atoms increases, weakening the binding force between atoms, the softening effect of the material becomes obvious. The PPB is broken and eliminated quickly at elevated temperature and high stress, replaced by dynamic recrystallized grains, and good metallurgical bonding between the powder particles is achieved [20]. No PPB and macroscopic cracks are observed in PA1 and MA1-36h alloy consolidated at 1050 °C.

The formation of crack in hot compression are described by the three steps in the schematic, as shown in Figure 6. At initial stages, loose powder began to slide and rearrangement under low pressure. Thereafter, as the stress increase, the plastic deformation of powder gradually happens, when the stress reaches the yield point of the powder. As hot compress continue, strain increases gradually, dynamic recrystallization (DRX) occurs. However, when hot compression temperature is low, PPB can hardly be broken, and gradually is stretched into ellipsoid as strain increases [20], leading to the concentration of stress at PPB, microcracks generates and expands along the PPB, leading to the formation of macroscopic cracks eventually.

### 3.3. XRD of the Powder and Alloys Analysis

Figure 7a shows the XRD patterns of precursor and ball milling powder. γ-Ni and YH_2_ diffraction peaks are visible in the mixture powder of PA and YH_2_ powder. After ball milling 36 h, only γ-Ni diffraction peak are observed in MA powder, YH_2_ is unstable and dissociated to Y and H during the MA, which is consistent with other studies [21]. Meanwhile, the impact energy generated by the ball milling results in a reduction of grain size and an accumulation of the lattice strain, which leads to the decrease of diffraction peak intensity, increase of width and angular deviation [22]. The XRD patterns of PA1 and MA1-36h alloy consolidated at 1050 °C/0.1 s^−1^ is illustrated in Figure 7b, the diffraction peaks of Y_4_Al_2_O_9_ and TiO_2_ are detected in MA1-36h alloy, only the diffraction of TiO_2_ can be found in PA1 alloy.

### 3.4. Microstructure Characterization

The grains at center regions of PA1 and MA1-36h alloy consolidated at different conditions are presented by EBSD IPF, as shown in Figure 8 and Figure 9. Generally, the samples consolidated at higher temperatures and lower strain rates have large average grain size. Significant grain growth happens when the temperature excesses 950 °C, especially at low strain rate. As the consolidation temperature increases from 850 °C to 1150 °C at strain rate of 0.1 s^−1^, the average grain size of PA1 alloy increases from 2.76 µm to 16.5 µm, and that of the MA1-36h alloy increases from 0.18 µm to 0.31 µm. The grain evolution is generally controlled by dynamic recovery (DRV), DRX, and grain growth processes [23,24,25,26]. For the PA1 alloy, comparing with the initial microstructure of precursor powder of PA1 (Figure 2b), grains are not uniform and some large grains only deform under stress rates of 0.1 s^−1^ and 1 s^−1^ at 850 °C. Grains of alloy consolidated at 950 °C are more uniform than that of the samples consolidated at 850 °C, with the increase of thermal activation. Grain growth plays a dominating role during the microstructure evolution when the temperature excesses 950 °C. Comparing with microstructure of the PA1 alloys, the grains of MA1-36h alloys consolidated at 850–950 °C keep fine during consolidation, and the average grain size fluctuates slightly. The grains grow gradually in samples as the consolidation temperature increasing to 1050 °C. Moreover, it is clear that the grains decrease from micron to submicron by comparing PA1 alloy with MA1-36 alloy, the grains are refined by ball milling. The grain growth rate of PA1 alloy is more rapidly than that of MA1-36h alloy with the temperature above 1050 °C, mechanism behind that will be discussed thereafter. In terms of strain rate, the strain rate has ambiguous effects on the grain evolution at 850–950 °C, as the consolidation temperature excesses 1050 °C, the average grain size of samples consolidated at 5 s^−1^ is smaller than that of other strain rates, and at a certain temperature, grains in alloys deformed at 5 s^−1^ are more uniform than these of the counterparts consolidated at other strain rates. Accordingly, the consolidation temperature of near 1050 °C with strain rate of 5 s^−1^ is suggested, to obtain homogeneous distributed fine-grains.

Figure 10 shows the grain misorientations of PA1 and MA1-36h alloys consolidated at different conditions. The influence of consolidation conditions on the grain evolution can be reflected by grain boundary misorientation [27]. The dislocation occurs in the process of consolidation, which leads to the misorientation in grains, and forming low angle grain boundaries (LAGBs, <15°). The higher the dislocation density, the larger the proportion of LAGBs. DRX is triggered, when the dislocation density reaches critical dislocation density. The dislocation is absorbed during the grain growth, resulting in the decrease of dislocation density, the fraction of LAGBs decreases. With sufficient driving force and time, DRX continues to occur and the dislocation density decreases further. For the PA1 alloy, comparing with PA1 powder (Figure 2), the average grain misorientation of samples decreases, as shown in Figure 10a. Generally, As the consolidation temperature increases from 850 °C to 1050 °C, the thermal activation increases, the fraction of high angle grain boundary (HAGBs, >15°) and the average grain misorientation of samples increases. However, the average grain misorientation decreases at 1150 °C, it is speculated that the grain growth leads to the decrease of the grain quantity in the same area. For the MA1-36h alloy, the average grain misorientation of samples increases with the increase of temperature except 1150 °C/1 s^−1^ (Figure 10b).

In addition, Figure 11 shows the TEM bright field images of PA1 and MA1-36h alloy consolidated at 1150 °C/5 s^−^^1^. Figure 11a shows a large-grain region with a high dislocation density in the PA1 alloy. The dislocations have long, relatively straight-line segments, with sharp corners and a serrated appearance in the other locations. Figure 11b illustrates the microstructure of MA1-36h alloy, significant nanoparticles and dislocations are noticed in sample, the dislocations interact with the nanoparticles, indicating that they are strongly pinned by nanoparticles. The nanoparticles are distributed in grain and grain boundary, ranging in size from a few nanometers to tens of nanometers.

The grain growth rate of MA1-36h alloy is retarded comparing with PA1 alloy. Since nanoparticles detected as Y_4_Al_2_O_9_ by XRD are dispersed in MA1-36h alloy, which prevent DRV and grain growth via pinning dislocations and grain boundaries [28], the grain boundaries of MA1-36 h alloy are more stable than that of PA1 alloy, even at very high temperature [29].

The local misorientation images of PA1alloy and MA1-36h alloy consolidated at different conditions illustrate the hardening status of grains, as presented in Figure 12. In general, the specimens consolidated at higher temperature shows lower local misorientation. The strain hardening of PA1 consolidated at 850 °C is significant, as shown in Figure 12a, which is related to dislocation accumulation in materials. With the increase of temperature, the strain hardening gets relieved by DRV at higher temperature, which also facilitates the DRX via subgrain formation [30]. In addition, as the strain rate increases from 0.1 s^−1^ to 5 s^−1^, the PA1 alloy presents lower local misorientation except at 1050 °C. However, the strain rate has ambiguous effects on the local misorientation evolution in MA1-36h alloy, as shown in Figure 12b. 

EBSD maps in Figure 13 and Figure 14 depict the deformed, substructured and recrystallized grains in the specimens consolidated at different conditions. In general, with the increase of consolidation temperature, the frequency of deformed grains deduces gradually, while fractions of the recrystallized grains get larger at the same strain rates. In general, the deformed grains of PA1 alloy decreases gradually with consolidation strain rate increasing from 0.1 s^−1^ to 5 s^−1^. Astonishingly, the substructured grains in MA1-36h alloy consolidated at strain rates of 1 s^−1^ are more than the counterparts consolidated at strain rates of 0.1 s^−1^ and 5 s^−1^.

### 3.5. The Variation of Hardness

The hardness of specimens manufactured at different conditions is shown in Figure 15. The hardness of PA alloy and MA alloy ranges in 154.8–344.9 HV and 421.1–697.3 HV, respectively. As expected, the hardness of MA alloy is significantly higher than that of PA alloy, and the hardness increases with the ball milling time prolonging. In addition, the hardness of samples is also strongly dependent upon temperature. The hardness of samples decreases with the increase of the consolidation temperature. For example, the hardness of PA1 alloy decreases from 288.6 HV to 166.3 HV, with the consolidation temperature increases from 850 °C to 1150 °C at strain rate of 5 s^−1^. The effect of strain rate on the alloy hardness is also presented. In general, the hardness of samples increases with increasing strain rate, especially at 1050–1150 °C. High deformation rate is conducive to the formation of high-density dislocation, higher energy storage and refine grain size, improving the properties of the alloy, which is consistent with the other literature reports [31,32,33,34,35,36,37,38].

## 4. Discussion

### 4.1. Microstructure Evolution during Consolidation

The grain evolution during thermal consolidation is generally controlled by DRV, DRX and grain growth [23,24,25,26]. The strain hardening is significant in specimens compressed at 850 °C, adequate energy is stored in materials by dislocation accumulation to trigger DRX, as illustrated by the local misorientation maps in Figure 12. As temperature increasing, the strain hardening gets relieved, the improved temperature enhances the formation of DRX nucleation via subgrain formation, the initiation of DRX is faster at higher temperature [30,39]. In general, the improved temperature increases the average grain size of samples based on Figure 8 and Figure 9. The improved temperature can also enhance the DRV and grain growth which relief the dislocation accumulation and consume the stored energy, making it less sufficient to trigger sufficient recrystallization.

In addition, in terms of strain rate, the grain growth is relatively slackened and average grain size drops during consolidation as the strain rate increases from 0.1 s^−1^ to 5 s^−1^, specially at 1050–1150 °C. The higher strain rate inhabits DRV and limits the time of boundary migration, DRX nucleation via coalescence of subgrain and the strain induced boundary migration is retarded [39,40].

### 4.2. Strengthening Mechanism of Nickel-Based ODS Superalloys

The hardness *H_V_* of samples can be divided into the matrix hardness *H*_0_, grain boundary strengthening *H_g_*, solid solution strengthening *H_ss_*, oxides strengthening *H_p_* and dislocation strengthening *H_d_*, it can be expressed by [41]: (1)$$H_{V}=H_{0}+H_{g}+H_{ss}+H_{P}+H_{d}$$

In this work, the difference of PA alloy and MA alloy can be estimated by equation:(2)$$∆H_{V}=∆H_{g}+∆H_{Ps}+∆H_{Dis}$$

The contribution of dislocation strengthening is given by Equations (3) and (4) [42,43]:(3)$$∆H_{d}=\alpha \mathrm{MGb}\sqrt{\rho_{d}}$$
(4)$$\rho_{d}=2\theta /\mu b$$
where *M* is Taylor factor equal to, *G* is shear modulus, *b* is Burgers vector, *α* is the dislocation strengthening coefficient, $\rho_{d}$ is the dislocation density, μ is the unit length, θ is the misorientation angle estimated by the local misorientation maps (Figure 12).

The grain boundary strengthening is related to the average grain size *D*, which can be determined by the Hall-Petch equation [44]: (5)$$∆H_{g}=\frac{k}{\sqrt{D}}$$
where *k* is the Halle-Petch strength constant. It is clear that the hardness of MA1-36 alloy is higher than that of PA1 alloy by comparing with the average grain size based on Figure 8 and Figure 9.

In addition, nano-oxides plays an important role in ODS superalloy, as shown in Figure 11. Based on Orowan strengthening mechanism [45,46], homogeneously dispersed oxide particles can suppress effectively dislocation movement, as well as grain boundaries migration [47,48]. Therefore, high hardness of MA1-36h alloy comes mainly from grain boundary strengthening and nano-oxides strengthening.

## 5. Conclusions

In summary, nickel-based superalloys with and without ODS have been fabricated by ball milling and efficient consolidation method. The relationship among the preparation process, microstructure and properties of alloy are studied. Basically, the following conclusions can be reached:(1)The PPB of nickel-based superalloys is difficult to be broken and eliminated at low temperature, which facilitates the crack nucleation and propagation during thermal consolidation. With increasing of temperature, the thermal activation of the material increases, the PPB is broken and eliminated quickly at elevated temperature and high stress.(2)The grain size is sensitive to the consolidation temperature, the average grain size increases with the increase of consolidation temperature. The average grain size of samples consolidation at strain rate 5 s^−1^ decreases and more uniform than low strain rate, since higher strain rate inhabits DRV and limits the time of boundary migration.(3)The hardness of nickel-based superalloys decreases with the increase of the consolidation temperature, strain rate, and the hardness increases after ball-milled for longer time. In addition, the hardness of nickel-based ODS superalloys is significantly higher than that of nickel-based superalloys without ODS due to the grain boundary strengthening and nano-oxides strengthening.(4)Basically, in order to obtain fine-grains, excellent properties and less cracking risk, consolidation temperature of near 1050 °C and strain rate of 5 s^−1^ are suggested.

## Figures and Tables

**Figure 1 materials-15-04087-f001:**
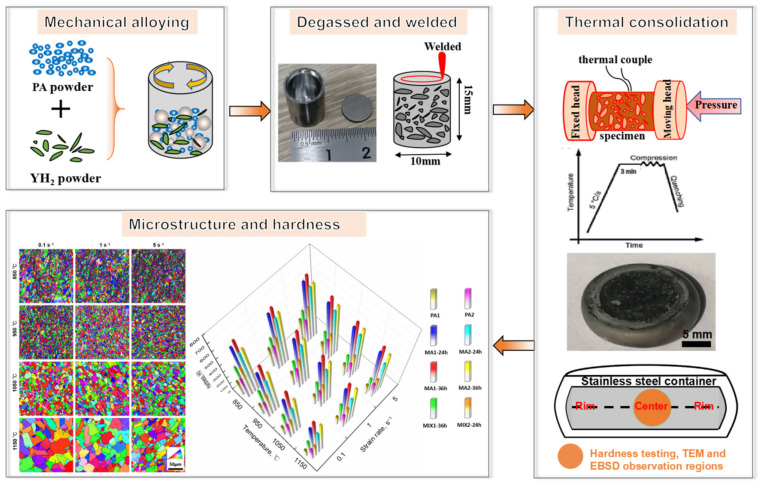
Flow-process diagram of samples preparation.

**Figure 2 materials-15-04087-f002:**
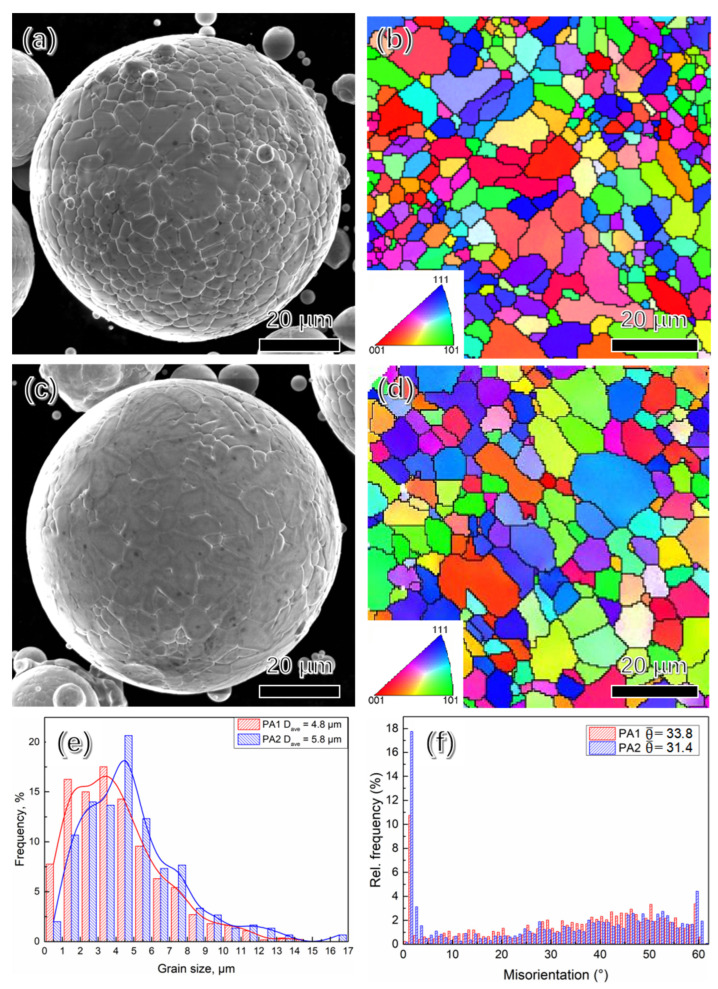
The SEM and EBSD IPF images of powders: (**a**,**b**) PA1; (**c**,**d**) PA2; (**e**) grain size distribution; (**f**) grain misorientation.

**Figure 3 materials-15-04087-f003:**
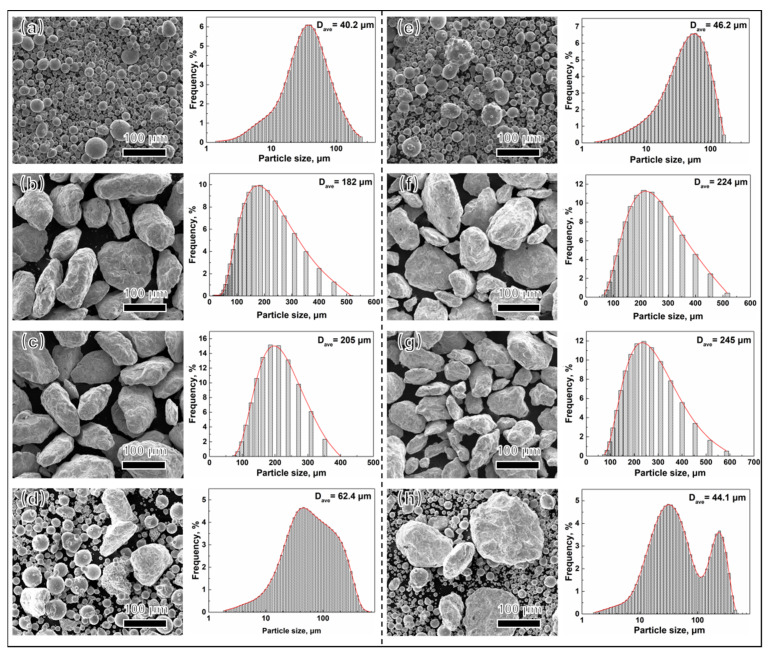
The microstructural feature and particle size distribution for (**a**) PA1, (**b**) MA1-24h, (**c**) MA1-36h, (**d**) MIX1-36h, (**e**) PA2, (**f**) MA2-24h, (**g**) MA2-36h, (**h**) MIX2-24h, respectively.

**Figure 4 materials-15-04087-f004:**
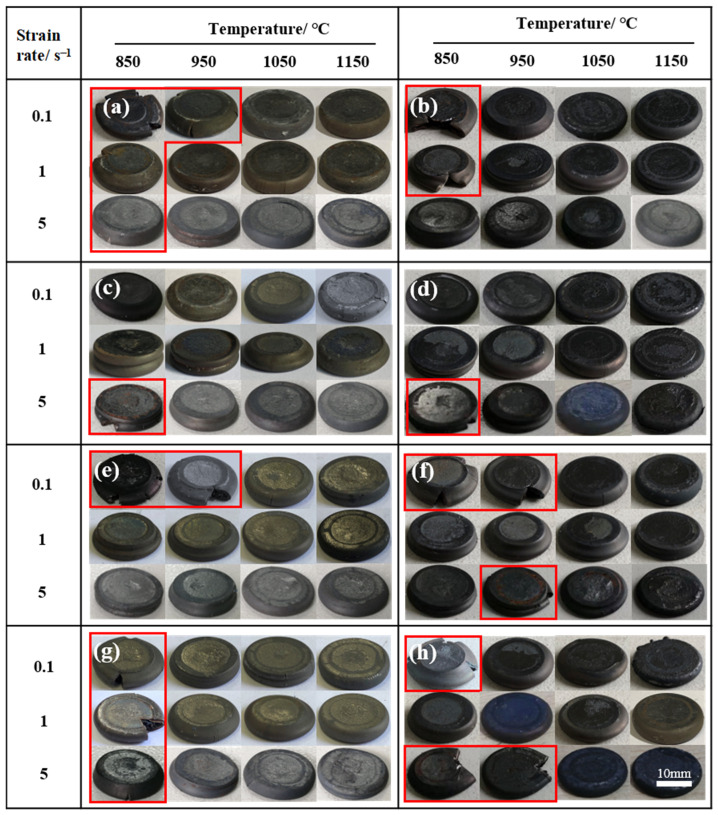
The shapes of specimens compressed at different conditions: (**a**) PA1; (**b**) PA2; (**c**) MA1-24 h; (**d**) MA2-24 h; (**e**) MA1-36 h; (**f**) MA2-36 h; (**g**) MIX1-36 h; (**h**) MIX2-24 h.

**Figure 5 materials-15-04087-f005:**
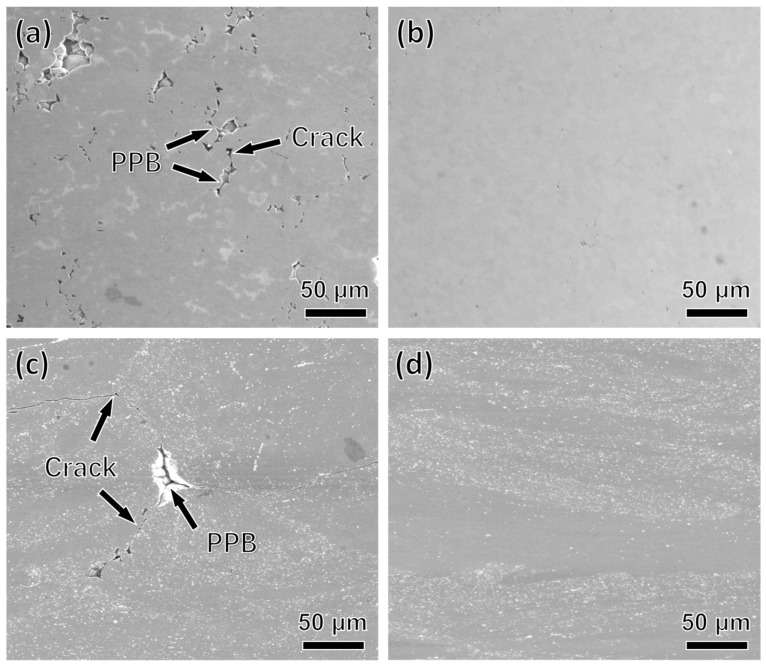
The SEM images of failure specimens consolidated at different temperatures with strain rate 0.1 s^−1^: (**a**) PA1: 850 °C; (**b**) PA1: 1050 °C; (**c**) MA1-36h: 850 °C; (**d**) MA1-36h: 1050 °C.

**Figure 6 materials-15-04087-f006:**
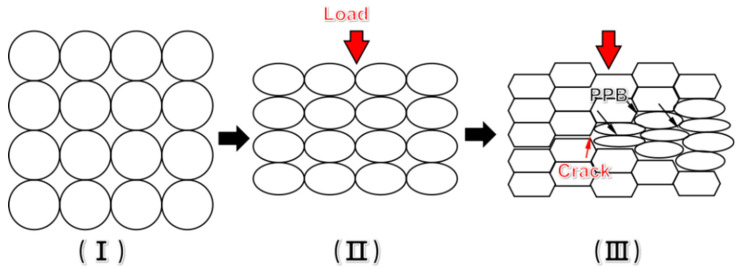
Schematic illustration of formation of compression cracks during hot compression. (**I**) prior to deformation, (**II**) deformation, (**III**) the formation of cracks.

**Figure 7 materials-15-04087-f007:**
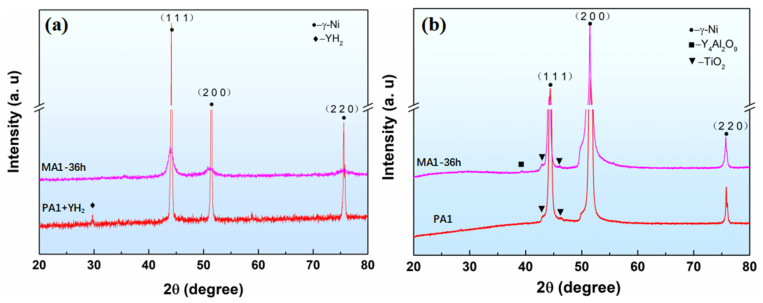
XRD patterns of (**a**) powders milled for different times and (**b**) samples consolidated at 1050 °C/0.1 s^−1^.

**Figure 8 materials-15-04087-f008:**
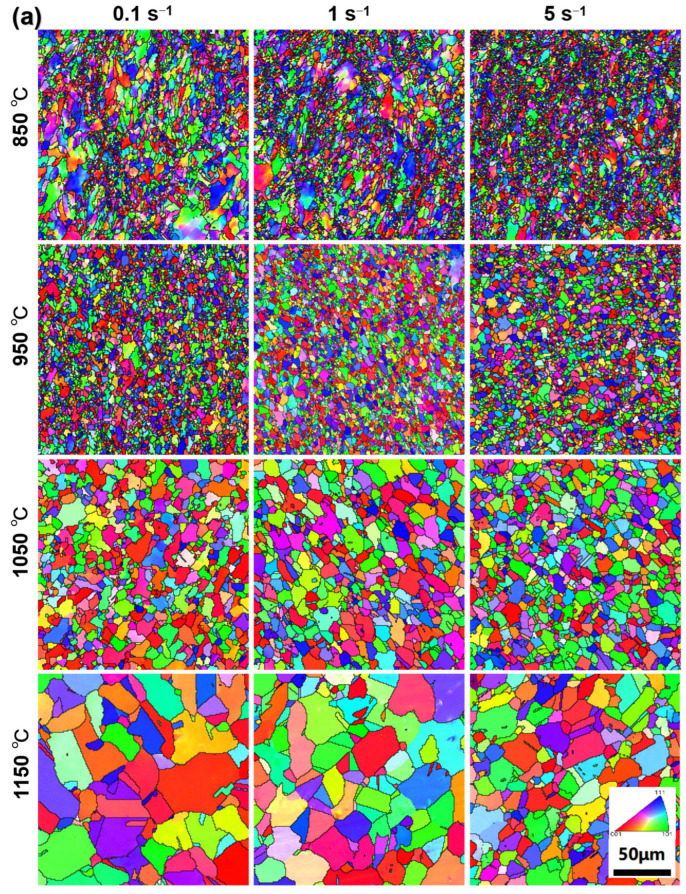
EBSD maps at center region of (**a**) PA1 and (**b**) MA1-36h alloys consolidated at different conditions.

**Figure 9 materials-15-04087-f009:**
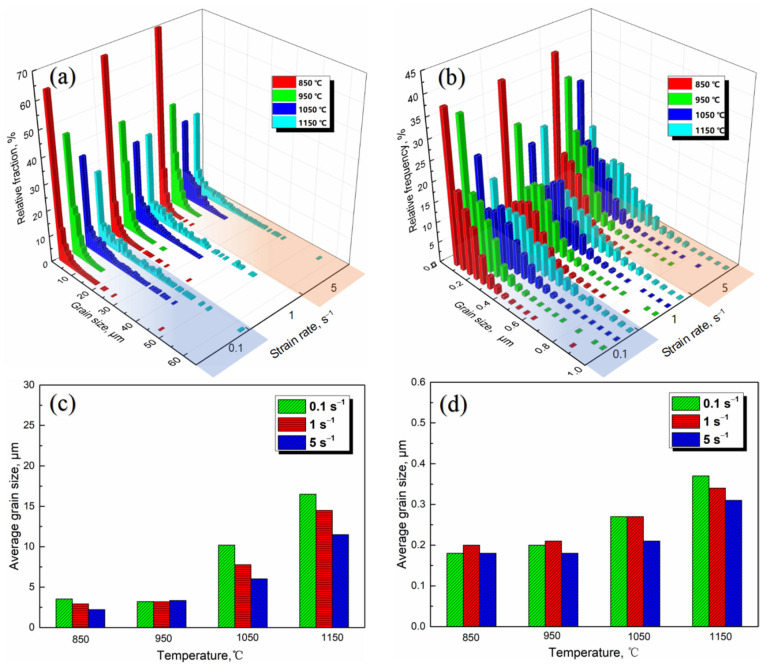
The equivalent grain size distributions and average grain sizes of (**a**,**c**) PA1 and (**b**,**d**) MA1-36h alloys consolidated at different conditions.

**Figure 10 materials-15-04087-f010:**
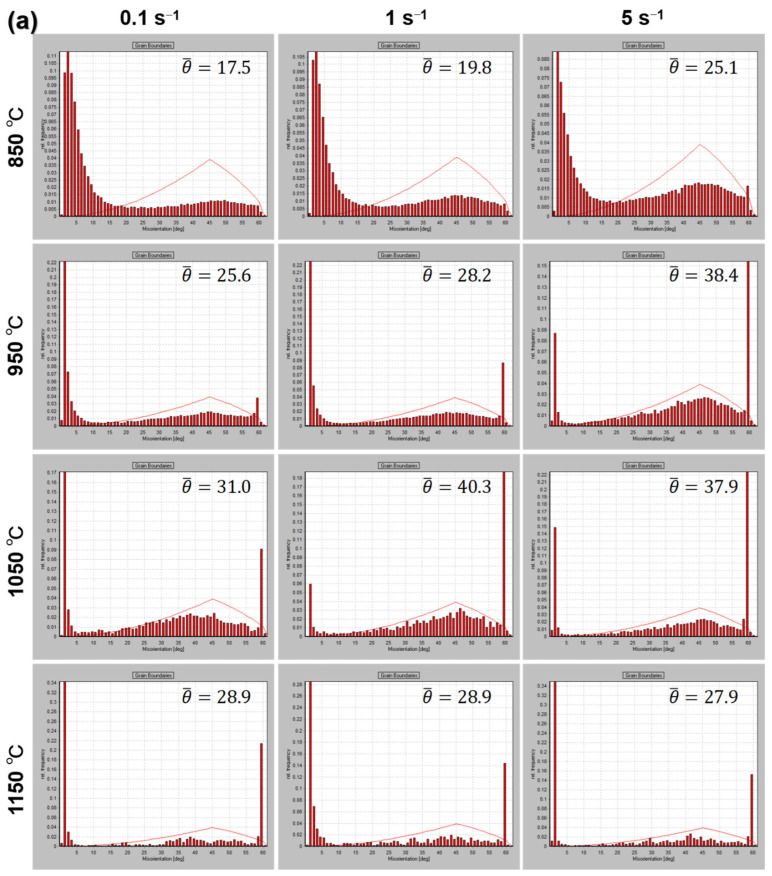
The grain misorientations of (**a**) PA1 and (**b**) MA1-36h alloys consolidated at different conditions.

**Figure 11 materials-15-04087-f011:**
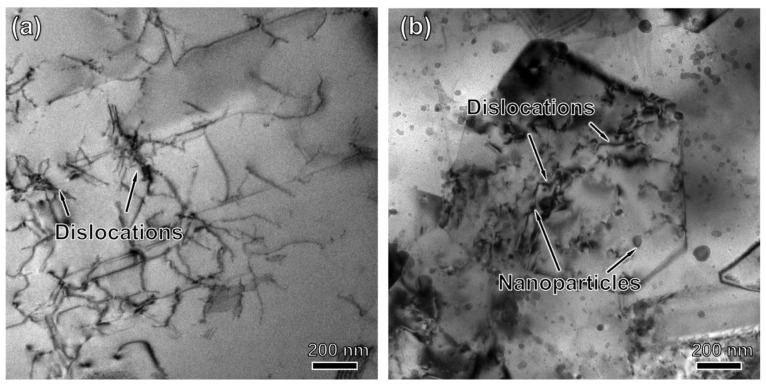
TEM photographs of (**a**) PA1 and (**b**) MA1-36h alloy compressed at 1150 °C/5 s^−1^.

**Figure 12 materials-15-04087-f012:**
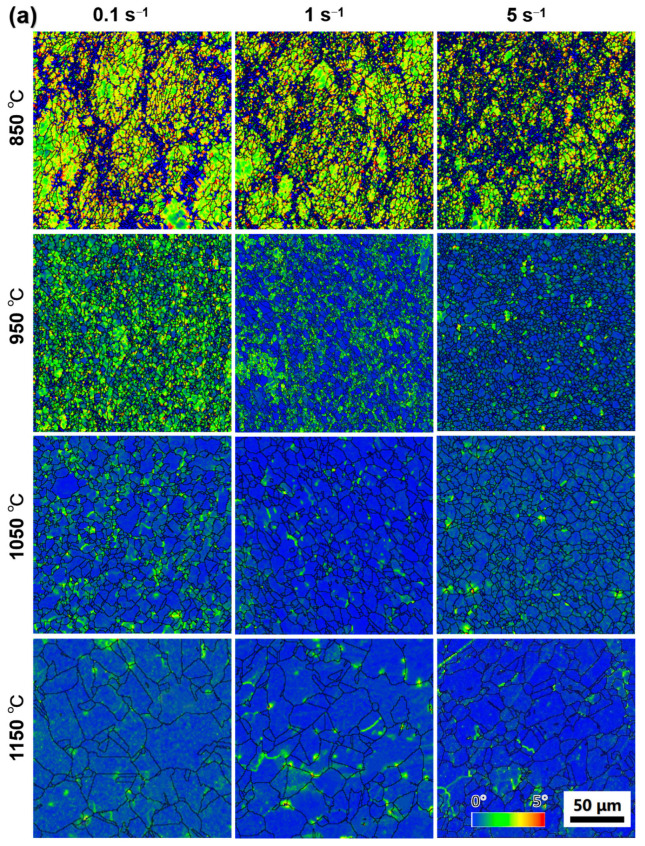
EBSD maps showing local misorientation in grains of (**a**) PA1 and (**b**) MA1-36h alloys consolidated at different conditions, wherein different colors ranging from 0° to 5° correspond to different misorientations as indicated by the color bar.

**Figure 13 materials-15-04087-f013:**
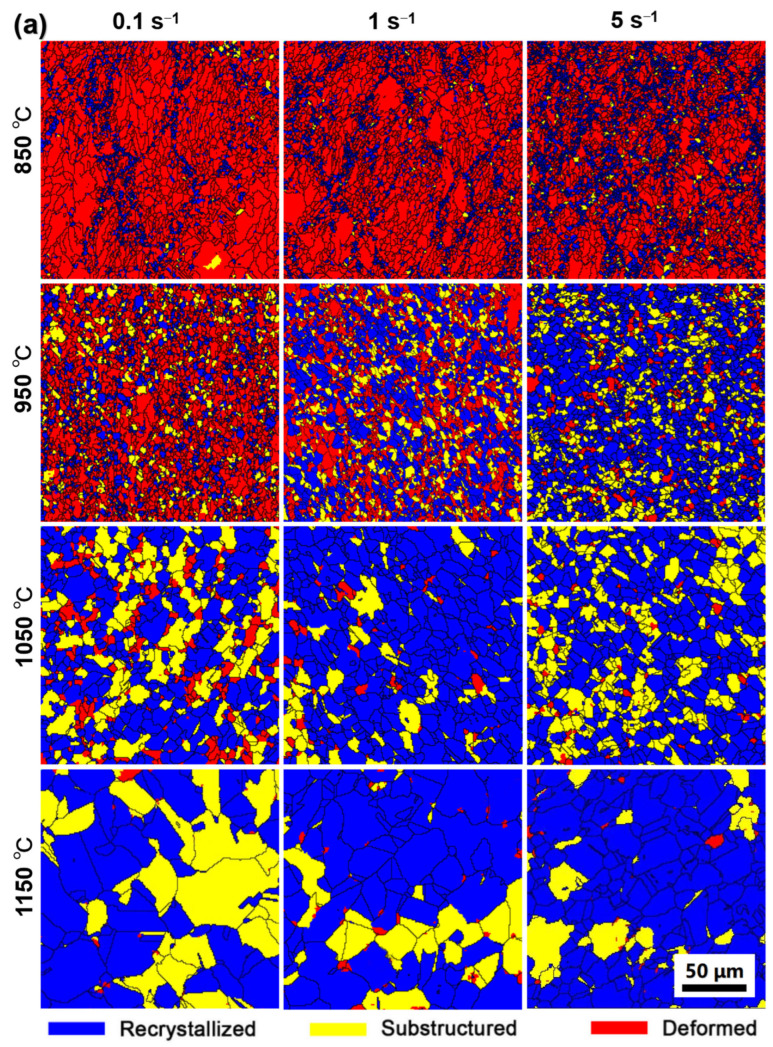
EBSD images showing the distributions of recrystallized, substructured, and deformed grains in (**a**) PA1 and (**b**) MA1-36h alloys consolidated at different conditions.

**Figure 14 materials-15-04087-f014:**
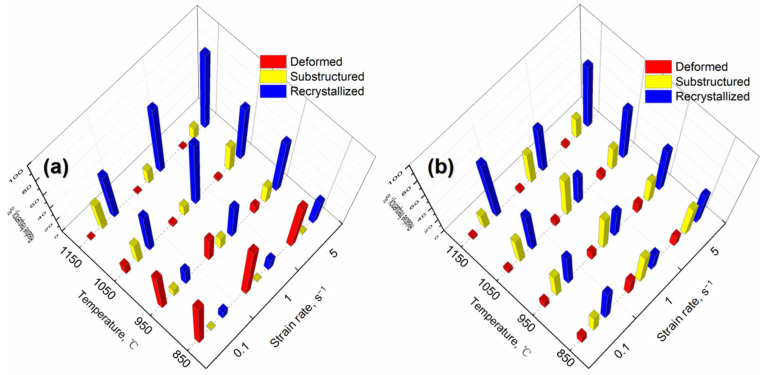
Relative frequency of deformed, substructured and recrystallized grains in (**a**) PA1 and (**b**) MA1-36h alloys consolidated at different conditions.

**Figure 15 materials-15-04087-f015:**
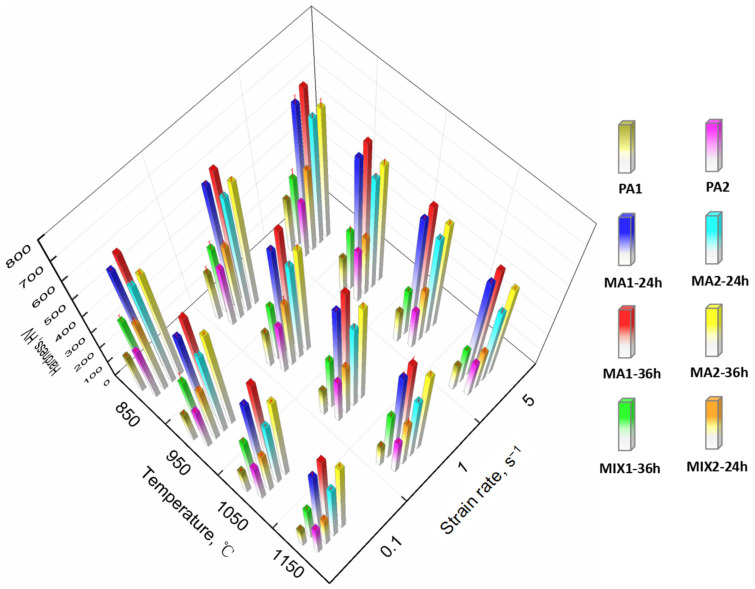
The hardness of alloys prepared at different ball milling time and consolidation process.

**Table 1 materials-15-04087-t001:** Chemical compositions of PA1 and PA2 powder in wt.%.

	Al	Cr	Fe	Ti	Y	C	Ni
PA1	0.25	20.5	0.67	0.56	0	0.054	Bal.
PA2	0.25	21	0.85	0.57	0.68	0.059	Bal.

**Table 2 materials-15-04087-t002:** Comparison of processing methods of specimens.

Specimen	Powder	Mill Time/h	Temperature/°C	Strain Rate/s^−1^
PA1	PA1 powder	0	850/950/ 1050/1150	0.1/1/5
MA1-24h	PA1 powder and 0.6%YH_2_ powder	24		
MA1-36h	PA1 powder and 0.6%YH_2_ powder	36		
MIX1-36h	MA1-36h powder and PA2 powder (ratio 1:2)	12 (mixed)		
PA2	PA2 powder	0		
MA2-24h	PA2 powder	24		
MA2-36h	PA2 powder	36		
MIX2-24h	MA2-24h powder and PA2 (ratio 1:2)	12 (mixed)		

## Data Availability

The data used to support the findings of this study are available from the corresponding author upon request.

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
