# Peer review of "Optimizing the Thermomechanical Process of Nickel-Based ODS Superalloys by an Efficient Method"

_materials, 2022, doi:10.3390/ma15124087_

Round 1

Reviewer 1 Report

This manuscript; beta; Examines. The position is very good. Good pictures are also provided. Unfortunately, proper analysis has not been done. Most of the manuscript is deafened by images that have no analytical support. There are no metallurgical studies. The slightest reference to the role of phases in the process is not used. Defects should be indicated by SEM images, not OMs. However, I suggest a major revision.

  1. Given that the handwritten results section is very comprehensive, more is expected from the introduction section. Please provide more details about the process as well as more studies that have been done in this field.
  2. Does not the nickel base superalloy used in this manuscript have a name? For example, IN525, IN718, etc.
  3. Please clarify what YH2 powder is and for what purpose it is added to the composition.
  4. A suitable analysis for crack formation is not provided in Figure 5. The authors do not mention the role of phases in creating crack formation. In addition, it is strongly recommended to prepare XRD‌ analysis powders to identify phases.
  5. In line 152, the authors refer to the creation of dynamic recrystallization. Please state the temperature of this event with a stronger reason.

Reviewer 2 Report

  1. Row 58, the author was used the 0.5 wt% alcohol for ball milling.
    What type of the alcohol? ethanol? methanol?

  2. Row 60, the author was adopted the "Mastersizer 3000".
    It should be revised to choice a univeral word.

  3. Row 69, thermal couples -> thermo-couples

  4. Row 239, the hardness value was indicated, but the unit is missing.

  5. It is interesting to study materials properties as functions of strain rates and temperature. 
    However, it is necessary to further consider what improvements have been made compared to the previous studies. etc, energy? time? cost?

  6. And, in what industies can it be applied?, how can it be effective?

  7. English correction and abbreviation making up is necessary

Round 2

Reviewer 1 Report

The manuscript has been improved somewhat well.
- Why are some images repeated twice?
- EBSD‌ images are well displayed and still do not provide good resolution. For example, what effect has recrystallization had on the angle of the boundaries? What is the angle of the boundary before and after recrystallization? Does this angle affect the mechanical properties?
- If possible, use the SEM‌ image to show the crack in Figure 5. The cracks in the light images in Figure 5 do not show where they occurred.

Author Response

Dear Materials’ Reviewer,

We have fully revised the manuscript according to your comments and suggestions, and the revised parts have been marked in red. We believe that our revised manuscript has addressed the concerns from your. Our responses are listed as following:

1. Why are some images repeated twice?

Our Response to Reviewer Comment: We have rechecked the images and no images were found to be repeated twice. If some photos are similar in Fig .4, which is due to the small size of images, and some details are indistinguishable. Please point out if you think those images are repeated, and we'll check them again.  

2. EBSD‌ images are well displayed and still do not provide good resolution. For example, what effect has recrystallization had on the angle of the boundaries? What is the angle of the boundary before and after recrystallization? Does this angle affect the mechanical properties?

Our Response to Reviewer Comment: The grain misorientation images of powder and samples were added in the article, the angle evolution of the boundaries was analyzed, Generally, combined with vickers hardness, the smaller the average grain misorientation of samples, the higher the vickers hardness.

3. If possible, use the SEM‌ image to show the crack in Figure 5. The cracks in the light images in Figure 5 do not show where they occurred.

Our Response to Reviewer Comment: The light images of samples were replaecd by SEM‌ images in Figure 5.

Thanks for your consideration!

Yours sincerely,

Corresponding authors: Pan Chen, Baogang Liu

Central South University

Changsha, Hunan Province P. R. China 410083

Tel: (+86)0731-88830938

E-mail: Panchen@csu.edu.cn (P. Chen); liudd2016@126.com (B. Tan).

Reviewer 2 Report

All comments have been properly corrected.

This paper is possible to publish in this journal.

Author Response

Dear Materials’ Reviewer,

Thank you for your suggestion!

Yours sincerely,

Corresponding authors: Pan Chen, Baogang Liu

Central South University

Changsha, Hunan Province P. R. China 410083

Tel: (+86)0731-88830938

E-mail: Panchen@csu.edu.cn (P. Chen); liudd2016@126.com (B. Tan).